# Massively Parallel CRISPR-Cas9 Knockout Screening in Sheep Granulosa Cells for FSH Response Genes

**DOI:** 10.3390/ani14060898

**Published:** 2024-03-14

**Authors:** Zaixia Liu, Lingli Dai, Tianhao Sun, Yongbin Liu, Yanchun Bao, Mingjuan Gu, Shaoyin Fu, Xiaolong He, Caixia Shi, Yu Wang, Lili Guo, Le Zhou, Fengying Ma, Risu Na, Wenguang Zhang

**Affiliations:** 1College of Animal Science, Inner Mongolia Agricultural University, Hohhot 010018, China; 1181703365@emails.imau.edu.cn (Z.L.); lingli20022006@sina.com (L.D.); byc@emails.imau.edu.cn (Y.B.); mingjuangu@imau.edu.cn (M.G.); caixiashi@imau.edu.cn (C.S.); zl2022@emails.imau.edu.cn (L.Z.); mafengying@emails.imau.edu.cn (F.M.); 2Inner Mongolia Engineering Research Center of Genomic Big Data for Agriculture, Hohhot 010018, China; 13474912747@163.com; 3Veterinary Research Institute, Inner Mongolia Academy of Agricultural and Animal Husbandry Sciences, Hohhot 010031, China; 4School of Life Science, Inner Mongolia University, Hohhot 010021, China; 15704715111@163.com (T.S.); ybliu@imu.edu.cn (Y.L.); 5Institute of Animal Husbandry, Inner Mongolia Academy of Agricultural and Animal Husbandry Sciences, Hohhot 010031, China; fsy@imaaahs.ac.cn (S.F.); xl@imaaahs.ac.cn (X.H.); 6College of Veterinary Medicine, Inner Mongolia Agricultural University, Hohhot 010018, China; wangyu@imau.edu.cn; 7College of Life Science, Inner Mongolia Agricultural University, Hohhot 010018, China

**Keywords:** granulosa cells, FSH, massively parallel CRISPR-Cas9 library, dose-dependent genes

## Abstract

**Simple Summary:**

The development of ovarian follicles is mainly regulated by the follicle-stimulating hormone (FSH) released by the pituitary gland. FSH acts on the granulosa cell receptor *FSHR* to regulate oocyte development, follicular maturation, and ovulation. In this study, the optimal dose of FSH for promoting granulosa cells was explored by combining massively parallel sheep genome CRISPR-Cas9 knockout screening and transcriptome analysis. The CRISPR-Cas9 knockout libraries were designed on Chr 2, 3 and X and the sheep granulosa cell libraries were knocked out. This may be the first time that two methods have been combined to identify new FSH-responsive genes and pathways not foreseen in previous studies.

**Abstract:**

Follicle-stimulating hormone (FSH) regulates ovarian follicle development through specific gene expression programs. Granulosa cells (GCs) are somatic cells surrounding the oocytes, secreting gonadotropins to regulate ovulation and promote follicular development. By analyzing the effects of different doses of FSH on the proliferation of GCs, we found that adding 10 ng/mL of FSH, as the optimal concentration, could promote the growth of GCs. Furthermore, we have successfully constructed the first CRISPR-Cas9 knockout library targeting the genes on chromosomes 2 and 3 and the X chromosomes of the sheep massively parallel coding gene, as well as an ovarian GCs knockout cell library. For the first time, we have exposed the knockout cell library to a concentration of 10 ng/mL FSH to explore the underlying mechanisms. Through this screening, we have identified 836 positive–negative screening genes that are responsive to FSH, thereby revealing the regulatory mechanisms and screening the functionality of candidate genes. Next, RNA-Seq of control (0 ng/mL), low (10 ng/mL), and high (100 ng/mL) doses of FSH revealed 1708 differentially expressed genes, and combined with 836 genes, we obtained 129 FSH dose-dependent genes with extremely significant differences. This enables us to delve deeper into investigating and identifying the mechanisms by which FSH regulates GCs. More generally, we have discovered new regulatory factors and identified reproductivity-associated major effectors. These findings provide novel research directions for further studies on sheep reproduction.

## 1. Introduction

According to previous studies, sheep ovarian tissue expresses 85.5% of all genes expressed, expresses more complex transcriptomes and more genes, and participates in both BMP receptor signaling and immune response [1]. In the process of sheep reproduction, the ovaries regulate oocyte development, follicle maturation, and ovulation by releasing sex hormones. The development of ovarian follicles is mainly regulated by FSH (follicle-stimulating hormone), and LH (luteinizing hormone) released by the pituitary gland. FSH initiates three key events in ovarian follicles: proliferation of granulosa cells (GCs), estrogen synthesis, and expression of LHGCGR in GCs [2,3,4]. These events are critical for the follicular development to the preovulatory stage and oocyte maturation. GCs are the structural components of follicles, which are essential for the control of oocyte growth, meiosis, cytoplasmic maturation, and transcriptional activity in oocytes [5]. FSHR (follicle-stimulating hormone receptor) is located on chromosome 3 of sheep and plays an important role in GCs by binding to FSH. LH and FSH are transmitted to the ovaries through blood circulation, and LH acts on the luteinizing hormone (LH) receptor (LHCGR) to stimulate androgen production; FSH acts on GCs’ FSHRs to regulate androgen conversion to estrogen, causing a spike in LH production from mature follicles which induces rupture and promotes the release of mature oocytes [6]. When FSH and LH reach a certain proportion in the blood, ovulation is triggered, which helps to accurately assess ovarian reserve, ovarian response, and ovulation ability [7].

The FSH hormone plays an important role in the selection and growth of dominant follicles, which has a positive effect on improving the lambing rate and multiple birth performance of sheep. FSH fluctuation is consistent with follicular growth waves and fluctuates depending on the number and size of follicles developing. Oocytes stop growing when the fetus is in the first meiotic diplotene phase of life, and ovarian GCs and theca cells are key factors in oocyte maturation [8]. FSH regulates follicle development through a specific gene expression program and is widely used in assisted reproductive technology. FSH treatment alone can increase FSH binding capacity and FSHR mRNA levels and promote oocyte development, follicular maturation and ovulation [9]. Studies have shown that the optimal dose needs to be determined when adding FSH, as too high a dose may lead to overstimulation, multiple pregnancies, and serious side effects, such as ovarian hyperstimulation syndrome; meanwhile, too low a dose may lead to poor ovarian response or pregnancy failure [10]. At present, although a small group of genes regulated by FSH in GCs have been investigated, only a few studies have analyzed FSH-regulated genes at the transcriptome level. Therefore, in the present study, we systematically investigated the expression of genes regulated by the addition of FSH hormone when culturing sheep primary GCs using a suitable in vitro cell culture model and further analyzed the transcriptional changes in GCs regulated by FSH during follicular development. We did so to further decipher the FSH signaling mechanism by studying the regulation of gene expression by FSH at the transcriptome level, to better understand the molecular basis of follicular development, and to develop reproduction-related biomarkers.

Gene editing is a type of prospective gene mutation that is achieved by inserting, deleting, modifying, or replacing DNA in the genome of an organism. DNA constructs carrying homologous extensions are integrated into the target site of the genome using endogenous homologous recombination mechanisms within the cell [11]. CRISPR-Cas9 will not degrade immediately after cutting and repairing the genome of the target site and may continue to cut off the target site until it is completely degraded [12]. CRISPR-Cas9-mediated multiple gene editing can accurately target multiple genes and multiple traits at the same time, avoiding negative additive effects, and the technology has been applied in a variety of organisms. In mice, editing of EGFP and endogenous CRYGC genes in spermatogonial stem cells (SSCs) by the CRISPR-Cas9 system revealed no off-targeting, and gene-edited SSCs were transplanted in the varicocele of testes of infertile mice for spermatogenesis; when round sperm cells were formed, they were injected into mature oocytes to generate heterozygous offspring with corresponding mutant phenotypes [13]. In rats, the CRISPR-Cas9 system is used to make SSCs produce targeted germline mutations, and the EPSTI1 and ERBB3 genes are targeted for modification. The mutated spermatogonia are transplanted into the testis of the recipient rats for spermatogenesis, which can produce pure and non-chimeric mutant offspring [14]. In zebrafish, CRISPR-Cas9 technology was used for gene editing of oocytes. Compared with traditional methods, the efficiency of genome editing and germline transmission of in vitro fertilization in zebrafish was significantly improved, and heritable mutants were produced in zebrafish [15]. In sheep, CRISPR-Cas9 targets the key gene *PDX*1 of sheep oocytes. It was found that microinjection of CRISPR-Cas9 elements in stage II oocytes reduced cleavage, increased blastocyst rate, and targeted biallelic mutations compared with microinjection of zygotes with the same elements [16].

To optimize the efficiency and scope of knockout, we focused on chromosomes 2, 3, and X because these chromosomes have a large number of coding genes involved in follicle development, hormone regulation, and other key reproductive processes, such as the GATA4 gene located on chromosome 2, the *FSHR* and *LHCGR* gene located on chromosome 3, and *BMP*15 gene with X chromosome. We proposed that by selectively targeting the knockout of genes on chromosomes 2, 3, and X, we would be able to interfere with key genetic pathways involved in follicle development and hormone production, revealing novel insights into the molecular mechanisms of sheep reproductive biology. CRISPR-Cas9 technology was utilized to design and construct sheep massively parallel coding gene sgRNA, and the synthesized sgRNAs were cloned into the lentiviral vectors to generate the final CRISPR knockout libraries. Through the genome-wide CRISPR-Cas9 knockout library, a set of FSH-responsive genes that can be used to identify sheep granulosa cells was established, and the system was used to screen essential genes involved in sheep reproductive regulation. In the future, we will continue to use this system to identify genes that play regulatory roles in the reproductive signaling pathways of different pathogen-associated molecular patterns.

## 2. Materials and Methods

### 2.1. Isolation and Culture of Granulosa Cells

Sheep ovaries were obtained from a local slaughterhouse and placed in an incubator containing 1% penicillin–streptomycin solution (SP, Gibco, Carlsbad, CA, USA) in 37 °C normal saline before being transported back to the laboratory. The excess tissues and mesangium around the ovary were removed with sterilized tweezers and scissors. First, the tissues were washed three times with tap water and then quickly washed three times with 75% alcohol to remove blood and bacteria. Finally, the tissues were washed three times with normal saline. The follicles of about 3 mm were punctured with a 5 mL disposable syringe and the GCs were collected, transferred to a 15 mL centrifuge tube, and centrifuged at 1000 rpm for 5 min. After centrifugation, the supernatant was discarded and the cell sediment was resuspended by adding medium containing Medium 199 (M199) (Gibco, Carlsbad, CA, USA) +10% fetal bovine serum (FBS) (VivaCell, South America) +1% SP, and the washing of cell sediment was repeated three times. Then, the complete medium (M199 + 17% FBS + 1% SP) was added to the cell precipitate, inoculated in a cell culture flask (Corning), and cultured at 37 °C in a humidified atmosphere containing 5% CO_2_ cell culture incubator for 24 h. The cell growth was observed under a microscope to remove non-adherent cells and prevent cell contamination. Fresh culture medium was replaced with fresh culture solution every other day.

### 2.2. Effects of FSH on the Viability of Granulosa Cells

FSH hormone (Ningbo Second Hormone Factory, Ningbo City, Zhejiang Province, China) was configured into 2 μg/mL solution with a complete medium and then sterilized by 0.22 μm filter to obtain FSH medium. Then, the complete medium was used to configure FSH into different concentrations (0 ng/mL, 0.1 ng/mL, 1, 5 ng/mL, 10 ng/mL, 15 ng/mL, 20 ng/mL, 30 ng/mL, 50 ng/mL, and 100 ng/mL).

Briefly, the GCs were seeded into 96-well culture plates to produce 5 × 10^4^ viable cells in complete culture medium (M199 + 17% FBS + 1% SP) at 37 °C in a humidified atmosphere containing 5% CO_2_ until the cells’ confluence reached 80%. Then, we removed the medium and rinsed twice with PBS at 37 °C to establish a model of GCs treated with different concentrations of FSH. Eight replicate wells were set up for each concentration and cultured for 12 h, 1 d, 2 d, 3 d, 4 d, 5 d, 6 d and 7 d. Cell viability was detected by adding the CCK-8 reagent (Cell Proliferation and Cytotoxicity Assay Kit, Biosharp, Bangfu City, China) at different time points, and the absorbance at 450 nm was measured with a microplate reader (Bio-Rad, Hercules, CA, USA) at 1.5 h to determine the optimal concentration of FSH.

### 2.3. Genome-Wide sgRNA Library Design, Lentivirus Packaging, and Titration

A genome-wide CRISPR/Cas9 knockout library of sheep follicular granulosa cells was designed by selecting all protein-coding genes in sheep to identify genes related to FSH response. We transfected GCs with a genome-scale CRISPR knocked-out GeCKO V2.0 pooled library (refer to Zhang Lab- https://www.addgene.org/crispr/zhang/ for GeCKO v2.0 pooled libraries, accessed on 8 July 2021). The library contained 21134 encoding genes and the sgRNA library contained four different sgRNAs for each coding gene. The target gene locus of sgRNA contained PAM as NGG, and the length of sgRNA was 20 nt; the GC content ranged from 40% to 60%, and it was close to the initiation codon ATG. The genome-wide off-target evaluation of sgRNA investigated three base mismatches, which were unique in the genome and did not contain one and two base mismatch off-target sites, and the sgRNA with the least off-target sites. We used sgRNACas9 online software to design and find sgRNAs with high specificity as candidate sgRNAs, and 2000 sgRNAs without targets were used as a negative control [17]. Sheep genome-wide CRISPR-Cas9 sgRNA plasmid library construction and lentivirus packaging were carried out by Suzhou Hongxun Biotechnology Co., Ltd.

The construction method of the CRISPR-Cas9 library is as follows:(1)Design sgRNA sequence: ➀ Utilizing sgRNACas9 online software, we designed sgRNA sequences of protein-coding genes on the sheep chromosomes 2, 3, and X were, to search for sgRNAs with higher specificity to act as candidate sgRNA. ➁ The length of sgRNA was designed to be 20 nt.(2)Synthesis of sgRNA sequences: ➀ Synthesize sgRNA sequences with the Syno^®^ 3.0 technology platform. ➁ Perform PCR amplification on synthesized oligonucleotide probes using high-fidelity enzymes. ➂ Separate amplification products by 2% agarose gel electrophoresis and recover the desired bands.(3)Cloning into the vector: Clone the amplified product into the Lenti CRISPR.v2 lentiviral expression vector to form a linearized vector.(4)Construction of a library: ➀ Use Gibson Assembly^®^ MasterMix to spliced multiple DNA fragments together in one step. ➁ Ligated the recovered library fragment product with a linearized vector via Gibson Assembly^®^ MasterMix. ➂ Transformed the Gibson assembly product into Trans1 T1 competent cells. ➃ Extract the sgRNA plasmid library using plasmid extraction kit.(5)Packaging and transfection: ➀ Transfected the lentivirus-related packaging plasmid and the recombinant plasmid carrying the sgRNA library into 293T cells. ➁ The constructed sgRNA library vector was packaged into the lentivirus. ➂ Use the produced lentivirus to infect sheep granulosa cells for gene editing experiments.

### 2.4. Determination of the Optimal Concentration of Puromycin in Granulosa Cells

The granulosa cells were treated with puromycin at different concentrations. The steps are as follows: puromycin (Puro) was dissolved in sterile PBS to a working solution with a concentration of 1 mg/mL, filtered, and sterilized using a 0.22 μm Millex-GP filter before being stored at −20 °C for later use. The GCs were seeded in 6-well plates with 5 × 10^5^ cells and cultured for 24 h. The old medium was discarded and washed slightly with PBS. The antibiotic-free M199 medium containing 17 %FBS was used to prepare the culture medium with Puro concentrations of 0, 0.5 μg/mL, 1 μg/mL, 1.5 μg/mL, 2 μg/mL, and 3 μg/mL. We then replaced the fresh medium containing Puro once a day. We carried out a continuous culture for about 7 days, and when a certain concentration of cells had died, the lowest concentration of Puro was used as the optimal concentration for screening cells, and the concentration was used as a follow-up test. At a puromycin dose of 1 μg/mL, all cells died, and at 0.5 μg/mL, most cells survived. Therefore, a puromycin dose of 1 μg/mL was determined the optimal dose for screening cells after lentivirus infection.

### 2.5. Construction of the Massively Parallel CRISPR-Cas9 Knockout Cell Library of Granulosa Cells

The granulosa cells were inoculated into T25 culture flasks before lentivirus infection, cultured for 24 h, and infected with the library lentivirus at an MOI of 0.3 when the cells adhered to the wall and converged to about 80%. The cells were washed twice with PBS, the supernatant was discarded, 2 mL of serum-free M199 and polybrene with a final concentration of 8 μg/mL were added for incubation for 2 h, and the virus solution was diluted with complete medium for incubation for 24 h. After 3 days of infection, the fresh complete cell culture medium was replaced, and the medium was added without double antibody of Puro at a final concentration of 1 μg/mL. The medium was changed every day to observe its state. After screening with Puro medium for 7 days, some of the cells were collected to extract genomic DNA, and the sgRNA abundance was examined. The remaining cells were used for subsequent screening.

### 2.6. Massively Parallel CRISPR/Cas9 Screening of FSH Response Genes in Granulosa Cells

Granulosa cell knockout was performed on sheep chromosome 2, 3, and X libraries. After GCs were infected with each library virus, we divided the cell library into two groups. The knockout cell library obtained by only 1 μg/mL puromycin screening was the control group (without FSH stimulation), and the FSH stimulation group had 10 ng/mL FSH.

The above GC knockout cells were cultured in a complete medium containing 10 ng/mL FSH for 3 consecutive rounds, and 1 × 10^6^ cells were collected for genomic DNA extraction. The total genomic DNA was extracted using TaKaRa MiniBEST Universal Genomic DNA Extraction Kit Ver.5.0; for specific extraction steps, refer to the instructions.PCR was used to analyze the sgRNA information (Table 1). Briefly, 4 μg genomic DNA was taken and amplified with 2× Taq Plus Master Mix (Dye Plus) as a PCR enzyme. The total volume of the reaction was 50 μL and the specific reaction was as follows:



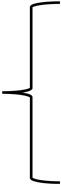

2 × Taq Plus Master Mix Π25 μL

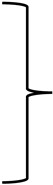

sgRNA2F2 μLsgRNA2R2 μLGeneme DNA4 μLAdd sterile enzyme-free water 50 μL

The PCR procedure is as follows:



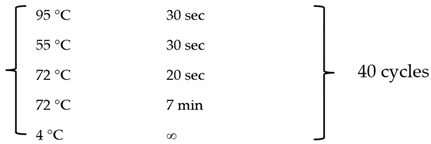



cWe carried out gel recovery and purification of the target fragment using the purification process according to the gel recovery kit.dThe purified product library was constructed for second-generation sequencing, and the second-generation sequencing was entrusted to the source of Novobio.eThe raw data after sequencing were subjected to quality control with FASTQC software Version 0.12.0, and the clean data after cleaning were obtained for subsequent analysis.fMAGeCK software (v1) [18] was used to compare the clean data with the sgRNA library sequence, and the sgRNA was counted and standardized. The MAGeCK screening method selects candidate genes with multiple enriched or depleted sgRNAs to reduce the possibility that the observed changes in sgRNA distribution are caused by the off-target activity of a single sgRNA. The R package MAGeCKFlute [19] was used for downstream analysis, statistical comparisons, and functional enrichment for positive and negative selection. Positive and negative selection were based on the phenotype of interest and available selection pressures for the screen. Positive selection exerts a certain screening pressure on the cell library that has successfully integrated sgRNAs, and the number of phenotype-related sgRNAs increases relative to the rest of the sgRNAs. Depending on the enrichment of sgRNA, genetic perturbations that produce screening phenotypes due to cell proliferation can be obtained and can screen resistance genes. Negative selection involves the depletion of phenotypic corresponding sgRNAs due to cell death, and the genes necessary for cell survival can be screened [20].

### 2.7. Granulosa Cells’ RNA-Seq and Bioinformatics Analyses under FSH Hormone Treatment

#### 2.7.1. RNA Extraction, Library Construction, and RNA-Seq

To better understand the biology and regulatory mechanisms of follicle development, we established different FSH models for culturing GCs. The GCs were seeded into 6-well cell plates (1 × 10^5^ viable cells) in a complete culture medium until the cell confluence reached 80%. Then, the medium was removed and washed twice using 37 °C PBS, and the GC model treated with FSH in control, low-, and high-concentration groups was established as follows: control group (Ctr: n = 3, 0 ng/mL FSH), low-dose group (Low, n = 3, 10 ng/mL FSH), high-dose group (High, n = 3, 100 ng/mL). The GCs were cultured with the above three kinds of medium for 48 h, the supernatant was discarded, and 1 mL RNAiso Plus (Takara Bio, Beijing City, China) was added. The cells were completely lysed by repeated blowing, and the cells were collected in a 1.8 mL cryopreservation tube and quickly frozen in liquid nitrogen. RNA-seq analysis was performed to comprehensively reveal the dynamic changes in GC gene expression induced by FSH and investigate the effects of FSH stimulation on GCs and ovarian follicle function. For total RNA extraction, quantity, quality, sequencing data alignment, and quantitative analysis, refer to Liu’s paper [1].

#### 2.7.2. Differential Expression Genes and Functional Annotation Analysis

The differential expression genes (DEGs) were assessed via the read counts for each sample using DESeq2 (version: 3.16) [21], and genes with a *p*-value < 0.05 and an absolute value of |log2 (Fold change)| > 0.583 (FC = 1.5) were used as DEGs.

Gene ontology (GO) and Kyoto Encyclopedia of Genes and Genomes (KEGG) enrichment analyses of DEGs were implemented using the g: profiler database (accessed on 8 January 2023) [22]. Enrichment terms with corrected adjusted *p*-values less than 0.05 were considered significantly enriched by DEGs.

### 2.8. Analysis of FSH Dose-Dependent Genes

To elucidate the expression characteristics of positive and negative screening genes and FSH dose-dependent genes in a massively CRISPR-Cas9 experiment, expression levels of positive and negative screening genes were matched with granulosa cell transcriptome data and the STEM program (https://www.cs.cmu.edu/~jernst/stem/; Accessed on 10 August 2023) for a series cluster analysis. Subsequently, the Pearson correlation coefficient between genes in each profile was calculated to screen out R > 0.8 and *p*-values < 0.05.

## 3. Results

### 3.1. Effects of FSH on the Viability of Granulosa Cells

In the culture of sheep with follicular GCs, different concentrations of FSH were added and cultured at different times. CCK-8 reagent was used to determine the cell proliferation rates at ten FSH concentrations (0 ng/mL, 0.1 ng/mL, 1, 5 ng/mL, 10 ng/mL, 15 ng/mL, 20 ng/mL, 30 ng/mL, 50 ng/mL, 100 ng/mL) and eight-time points (0 d, 0.5 d, 1 d, 2 d, 3 d, 4 d, 5 d, 6 d, and 7 d) (Figure 1). The results showed that with the increase in FSH treatment time, the growth density of GCs gradually increased, and the proliferation rate reached the highest point on the fourth and fifth days of culture. From the fifth day to the seventh day, due to the excessive cell convergence rate, partial death occurred, and the cell proliferation rate tended to decrease. By comparing the cell proliferation rate of 10 ng/mL, FSH was significantly higher than other concentrations (*p*-value < 0.05), indicating that an optimal dose of FSH could significantly improve the proliferation ability of GCs. Therefore, we selected 10 ng/mL FSH as the optimal dose of FSH hormone to be added to cultured granulosa cells.

### 3.2. Sheep CRISPR-Cas9 Massively Parallel Knockout Library Coverage and Homogeneity Testing

A massively parallel CRISPR-Cas9 knockout library was constructed for sheep chromosome 2, chromosome 3, and chromosome X. The CRISPR-Cas9 knockout library was sequenced for the next generation. The number of sgRNAs was 44,007,662 reads, corresponding to 16,457 sgRNAs (Appendix A). Compared with the originally designed synthetic sgRNAs, the coverage was 99.99%; the abundance was higher, and the sgRNA homogeneity was 6.6, meaning sgRNA library qualification standards were met (Table 2). The sequencing reads of sgRNA can be found in a relatively normal distribution, with high uniformity and low dispersion, and can be used for subsequent experiments (Appendix A).

### 3.3. Construction of Massively Parallel CRISPR-Cas9 Knockout Cell Library of Granulosa Cells

The target band of the knockout cell library was obvious at about 250 bp, which was relatively specific, there was no obvious non-specific amplification, and all were shown to be positive; meanwhile, the GCs in the blank group had no band and were negative (Figure 2).

Chromosome 2, chromosome 3, and chromosome X, which have relatively high gene density and are enriched in genes related to development, reproduction, and immune processes, were selected for GC knockout. After each library virus infected the GC cells, the knockout cell library GC-Cas9 was obtained after 1 μg/mL puromycin screening. When 10 ng/mL FSH was added to GC-Cas9 (GC-Cas9 + FSH), it was found that the cells in the FSH-added group grew faster than those in the non-added group, while all normal GCs in the blank control died (Figure 3).

### 3.4. Sheep Massively Parallel CRISPR-Cas9 Technology for Identifying FSH Responsive Genes in Granulosa Cells

#### 3.4.1. Knockout Cell Bank PCR Next-Generation Sequencing

The cell knockout library was divided into the control (chr2, chr3 and chrX) and the FSH addition group (chr2-FSH, chr3-FSH, and chrX -FSH). The genomic DNA of cells in each group was extracted, and the target fragments were recovered after purification for PCR next-generation sequencing. After quality control, the average clean reads were 8,619,589.667 bp, the average base number was 1.29 G, and the average Q30 was 92.73. The sequencing data were used for subsequent analysis (Appendix A).

The quality control data were mapped with a sheep sgRNA sequence library, and the average alignment rate was 39.14% (Figure 4). The target fragment size of the next-generation PCR double-ended sequencing was 250 bp, and the sgRNA sequence was close to one side of the fragment. The sequencing reads have 50% of the reads that cannot detect the sgRNA sequences, and the actual sgRNA sequence alignment rate was 39.14%, which was within the normal range.

#### 3.4.2. FSH Response-Related Gene Screening

The MAGeCK-RRA algorithm was used to evaluate the importance of genes by calculating the RRA score of the target genes’ loci corresponding to the sgRNAs; meanwhile, the target genes were ranked. Based on the criteria of log2 (fold change) > 2 and *p*-value < 0.05, positively selected target genes were obtained. The positively selected genes indicate that their knockout prolongs the life cycle of GCs under FSH treatment, which was the resistance gene of FSH. Negatively selected target genes were obtained based on log2 (fold change) < −2 and *p*-value < 0.05, indicating that GCs were sensitive to FSH treatment after knockout of these genes under the action of FSH, which is an essential gene for FSH action.

Further, the difference analysis was performed by using the number of sgRNA reads enriched in the cell library, and the target gene sequencing and functional enrichment analysis were performed on the screened positive and negative screening genes with significant differences. In total, 499 target genes were positively selected and 337 target genes were negatively selected in the three chromosomes (Figure 5A, Appendix A). Positively selected genes were mainly involved in metal ion transport, organic anion transport, and molecular adaptor activity. Negatively selected genes were involved in the monosaccharide biosynthetic process, small molecule biosynthetic process, sex differentiation, and tube morphogenesis. In the sex differentiation function, they were mainly associated with reproductive structure development, development of primary female sexual characteristics, gonad development, ovulation cycle, and various other reproductive functions. Positively and negatively selected genes were jointly involved in transcription factor binding, response to hormones, glutamatergic synapse and so on (Figure 5B). The KEGG pathways involved in positive and negative screening genes were the Glucagon signaling pathway and PI3K-Akt signaling pathway.

After CRISPR-Cas9 library positive selection, most cells died due to their sensitivity to FSH, while some cells survived due to resistance acquired through specific gene knockout. By detecting the gRNA carried by the surviving cells, genes associated with drug resistance can be identified. When positively selected genes are missing, GCs invoke FSH to promote cell growth, thus extending their life cycle. In contrast, when negatively selected genes are missing, these genes play a crucial role as necessary genes in granulosa cell growth or metabolic processes due to the sensitivity of GCs to FSH.

### 3.5. FSH-Regulated Granulosa Cells’ Transcriptome Analysis

#### 3.5.1. Gene Expression Analysis

Three levels of FSH were added to the medium of GCs in vitro, and GC samples were collected after 48 h of culture. After 48 h of culture, cell death occurred in the high-dose group (100 ng/mL FSH), while the number of cells in the low-dose group (10 ng/mL FSH) was greater than that in the control group (0 ng/mL FSH) and the high-dose group. The cell volume increased (Figure 6). GC samples were further collected for transcriptome sequencing, and Appendix A shows the concentration and integrity of RNA, in which 402,362,068 bp and 60.35 Gb raw data were obtained by transcriptome sequencing analysis (Appendix A). After data filtering, 78,775,454 bp, and 56.81 Gb clean data were obtained.

In all samples, a total of 18,508 genes were expressed (FPKM > 0.1), of which 17,248, 17,180, and 17,239 genes were expressed in the control group, low-dose group, and high-dose group, respectively (FPKM > 0.1). In addition, 16,071 genes were expressed in all samples and were considered to be widely expressed genes (Figure 7).

#### 3.5.2. Differential Expression Gene Analysis

The control, low-dose group and high-dose groups in GCs revealed a total of 363 DEGs in the high-dose group vs. low-dose group. There were 788 DEGs in the high-dose group vs. the control group, and 1148 DEGs in the low-dose group vs. the control group (Appendix A). The low-dose group with the addition of 10 ng/mL FSH served as the optimal dose for culturing GCs with the highest number of up-regulated genes. The addition of FSH resulted in an increase in the number of DEGs compared to no addition, while there were fewer DEGs in the high-dose group than in the low-dose group, suggesting that too high a dose of FSH increased apoptosis (Figure 8). These results revealed the expression patterns of different doses of FSH in granulosa cells.

#### 3.5.3. Systematic Function Analysis of Differentially Expressed Genes

To determine the possible functional significance of observed changes in gene expressed levels in the high- and low-FSH-treated GCs groups and the control group, a gene ontology (GO) term enrichment analysis was performed. The paper summarizes the significantly enriched GO terms of DEGs (Figure 9A,B). Interestingly, the DEGs were found to be significantly associated with cell development in the term enrichment of biological processes. The up-regulated genes are mainly involved in the functions of biological regulation, response to stimulus, immune system processes, and response to BMP. It is noteworthy that the TNF signaling pathway, PI3K-Akt signaling pathway, and ECM–receptor interaction were enriched among the up-regulated genes, suggesting that promoting cell development and cell differentiation might play a critical role in FSH-treated GCs (Table 3).

### 3.6. Screening and Analysis of FSH Dose-Dependent Genes

We performed a series cluster analysis on a comprehensive set of 836 genes that had been positively and negatively screened through massively parallel CRISPR-Cas9 experiments. To investigate the differential effects, we calculated the median expression levels for the control, low-dose and high-dose group of FSH treatment in RNA-Seq data. The Pearson correlation coefficient of the genes in each profile was calculated, and the screening criteria were R > 0.8 and *p*-value < 0.05. Notably, our analysis revealed three distinct patterns of gene expression in FSH-treated GCs. Specifically, profile 2 identified a decreased expression trend in the low-dose group (10 ng/mL FSH) and then stabilized expression in the high-dose group (100 ng/mL FSH). These genes are mainly involved in transferase activity, transferring phosphorus-containing groups, cellular component organization, and maintaining the stability of cell structure and function of the coordination function. In profile 0, 33 genes showed a trend of first decreasing expression and then rising expression, which involved to nitrogen compound metabolic process, primary metabolic process, catalytic complex, and other functions. Meanwhile, 65 genes in profile 6 showed a trend of first rising and then decreasing expression (Figure 10). This profile of genes is mainly involved in transcription factor binding, receptor complex and metal ion transport, and the cell surface receptor signaling pathway. These 129 significant FSH dose-dependent genes may jointly promote the maintenance of normal functions of cells and tissues, such as growth, development, reproduction and protein synthesis.

## 4. Discussion

The FSH hormone mainly acts on GCs in follicles and maintains the health of GCs by promoting the proliferation and differentiation of GCs and regulating the meiotic retardation and recovery ability of oocytes, thus becoming a key factor ensuring the survival of follicles [23,24]. Follicle maturation stimulates gene expression in GCs, activates many transcription factors, and inhibits transcription suppressor factors under the tight regulatory process initiated by FSH. FSH regulates the expression of at least 500 target genes in GCs, and the expression of these genes drives the development of follicles, enabling them to respond to the surge of LH and promote ovulation, oocyte maturation, and luteum formation. Thus, it supports embryo development after fertilization and implantation [25]. The goat single-cell RNA sequencing found that GCs vary during different development stages: *ASIP* and *ASPN* are highly expressed in early-stage GCs, while *INHA*, *INHBA*, *MFGE*8, and *HSD17B*1 are highly expressed in growing-stage GCs. Additionally, *IGFBP*2, *IGFBP*5, and *CYP11A*1 are highly expressed in GCs during the growth period [26]. Existing studies have demonstrated that FSH can regulate gene expression in granulosa cells through multiple mechanisms, including modifying H3K4me3 and inducing specific transcripts [27].

Nivet, A.-L., et al. [28] showed that short-term FSH stimulation inhibited the transformation of epithelial cells into mesenchymal cells, maintained follicles in the growth phase, and prevented differentiation. Regulating gene expression in vitro has physiological limitations, but it can help evaluate potential downstream responses. Through in vitro culture of GCs, this study found that an excess of FSH may induce GC death. As the concentration of FSH increases, the proliferation of GCs also increases, reaching a peak when the concentration of FSH is 10 ng/mL. This indicates that an appropriate dose of FSH has a significant impact on proliferation, apoptosis, and follicle formation. In poultry, after treatment with 50 ng/mL FSH and 200 ng/mL IGF1, cell proliferation or apoptosis-related genes increased, and the expression of caspase3 decreased, which stimulated the proliferation of granulosa cells and inhibited the apoptosis of granulosa cells [29]. Similar results were also found in this study. After adding FSH, the expression of the pro-apoptotic gene Casp3 (also known as Caspase3) in GCs was significantly decreased (*p* < 0.05). The expression levels of *BCL2L*1 and *BCL2L*13, the homologous genes of the anti-apoptotic gene BCL-2 family, were significantly increased (*p* < 0.05).

CRISPR-Cas9 is a precise and efficient gene-editing tool that can be rapidly and widely applied in researching gene function and improving economically important traits such as muscle quality, fiber length, coat color, and litter size [30]. Genome-wide CRISPR-Cas9 knockout technology has been applied to humans [31], mice [32], pigs [33], chickens [34], cattle other animals. Shalem et al. [35] performed genome-wide CRISPR-Cas9 knockout of 18,080 target genes and 64,751 gDNAs in humans to identify genes required for the growth and survival of cancer and pluripotent stem cells while screening out genes associated with vemurafenib resistance.

By further exploring the FSH-responsive genes in sheep granulosa cells, we utilized whole-genome CRISPR-Cas9 knockout technology to construct the knockout GC library and design unique sgRNA sequences of chromosome 2, 3, and X coding genes. During sheep CRISPR-Cas9 knockout screening, the addition of 10 ng/mL exogenous FSH hormone to GCs can significantly enhance the proliferation ability. Analysis of knockout cell libraries revealed that genes of chromosome 2 may be associated with cellular immune response, chromosome 3 may be related to reproduction, and genes on chromosome X may be linked to oxidative reactions and energy metabolism. Construction of the knockout cell library indicated that knockout or inactivation of certain essential genes related to growth may lead to cell death, suggesting that these genes may be the target genes for certain traits. Positive and negative selection were performed in sheep ovarian GCs to identify 836 genes for cell growth and target genes regulating GCs response to FSH, aiming to identify the genes related to reproductive processes such as follicle development, ovulation, and oocyte maturation.

Transcriptome analysis of GCs revealed 2999 DEGs, which were evenly distributed in different cellular compartments including the extracellular matrix. This observation implies that FSH regulates the differentiation process of granulosa cells by influencing multiple aspects such as protein, steroid biosynthesis, receptor signaling, and cell development. KEGG pathway analysis indicates that FSH appears to promote granulosa cell differentiation through the TNF signaling pathway, the PI3K-Akt signaling pathway, and ECM–receptor interaction. Massively parallel CRISPR-Cas9 and RNA-Seq screened genes were almost enriched in hormone response, cell growth and development. In the group with 10 ng/mL FSH, the expression levels of genes *EGR*1, *STAR*, *IGFBP*4, *WNT*6, *GLI*1, *TP*53, and *MITF* were significantly increased. However, as the FSH concentration increased, the expression levels of these genes gradually decreased or stabilized. Additionally, studies have shown that treatment with 1 ng/mL FSH for 8 h significantly increased the expression of *IGF*-2; the synergistic action of *IGF*-2 with FSH can enhance cell proliferation and the expression of *CYP19A*1, and blocking IGF-2 activity eliminates the stimulatory effect of FSH on *CYP19A*1 [36]. Another experiment showed that after 72 h of treatment with exogenous FSH, there was no difference in the number of GCs transfected with FSHβ siRNA or without FSHβ siRNA, while the number of GCs without FSHβ siRNA increased significantly after exogenous FSH treatment [37]. These results indicate the importance of ovarian FSH and that the effect of exogenous FSH on granulosa cells depends on ovarian FSH.

This study employed CRISPR-Cas9 whole-genome library lentivirus screening to identify genes related to FSH sensitivity and resistance. Next-generation sequencing technology was utilized to identify specific sgRNAs, and together with transcriptome analysis, FSH-responsive genes were screened. By constructing FSH-treated and untreated granulosa cells, we identified 129 genes that may be dependent on the dose of FSH in granulosa cells. We preliminarily screened functional genes related to FSH response in sheep follicular granulosa cells, laying the foundation for new research on the mechanisms of gonadotropin regulation in follicular development and making significant contributions to the discovery of new reproductive genes in sheep. In the future, we will establish a complete genome-wide CRISPR-Cas9 knockout library of ovarian GCs to explore more comprehensive genes related to sheep reproduction and other traits or disease resistance. In the meantime, more in-depth experiments are needed to elucidate the molecular mechanisms of these candidate genes.

## 5. Conclusions

In summary, FSH has a direct stimulatory effect on sheep GCs‘ proliferation and the capacity of CRISPR-Cas9 technology to obtain dose-dependent genes for FSH. Given its relevance to sheep production practices and its direct effect on lambing rates, the regulation of FSH in sheep ovarian tissue reproduction deserves more attention at the molecular level.

## Figures and Tables

**Figure 1 animals-14-00898-f001:**
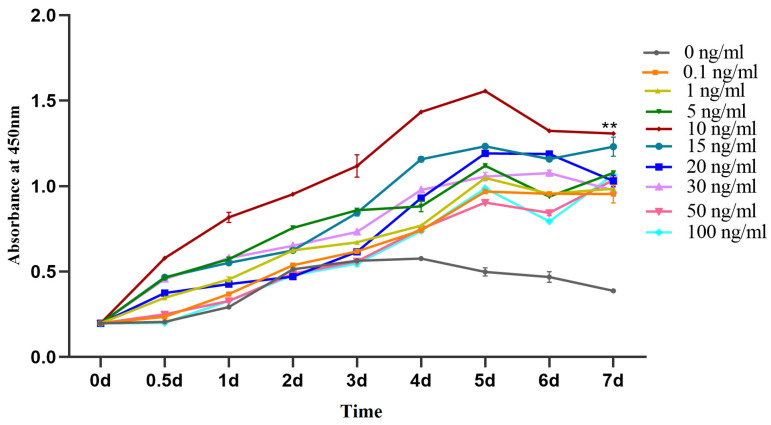
CCK8 assay was performed to assess the effect of FSH on granulosa cell proliferation. The X-axis indicates the number of days that GCs were cultured in vitro with different doses of FSH, and the Y-axis indicates the absorbance (OD value) at a wavelength of 450 nm, as detected by a microplate reader.

**Figure 2 animals-14-00898-f002:**
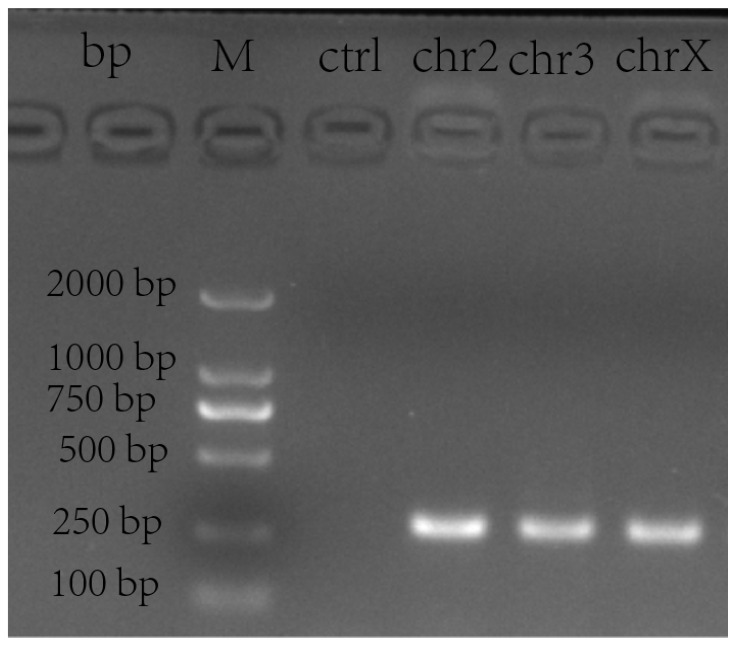
PCR amplification band position and specificity detection. The PCR amplification shows the length of bands in three chromosomes’ (chr2, chr3, chrX) knockout cell libraries and blank granulosa cells (ctrl).

**Figure 3 animals-14-00898-f003:**
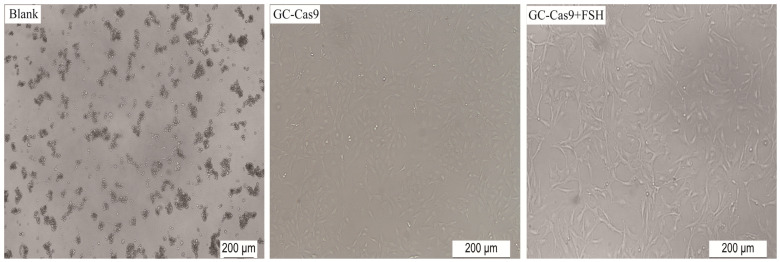
Microscopic examination of virus infection screening in the GC knockout cell library. Scale bars are 200 μm.

**Figure 4 animals-14-00898-f004:**
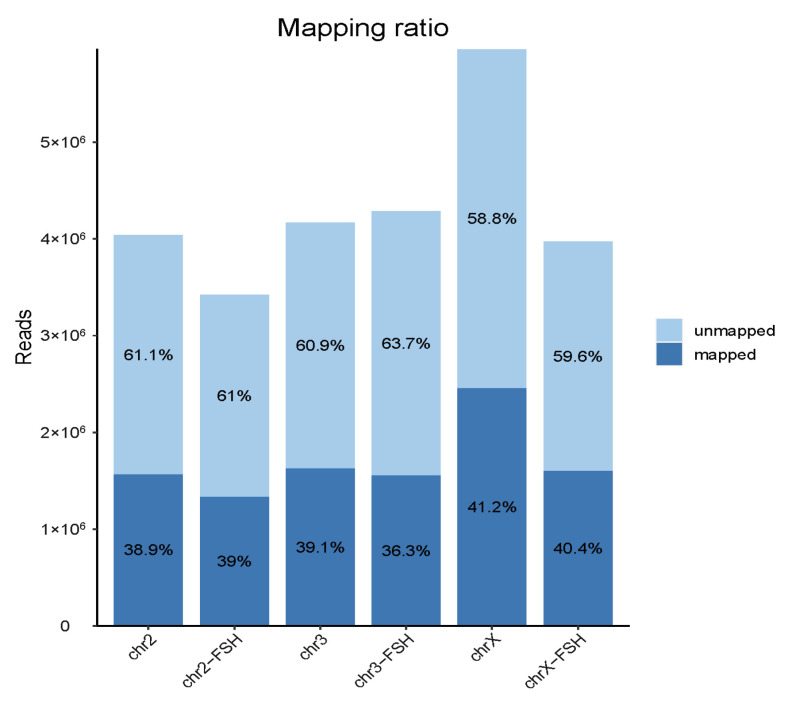
Read counts and mapping percentages of sgRNA sequencing in different treatment groups of GC genome knockout cells. chr2 was used as the initial cell library before screening the chromosome 2 virus library; chr2-FSH was used as the treatment group with 10 ng/mL FSH; chr3 was used as the initial cell library before screening the chromosome 3 virus library; chr3-FSH was used as the treatment group with 10 ng/mL FSH; chrX was used as the initial cell library before screening the chromosome X virus library; and chrX-FSH was used as the treatment group with 10 ng/mL FSH.

**Figure 5 animals-14-00898-f005:**
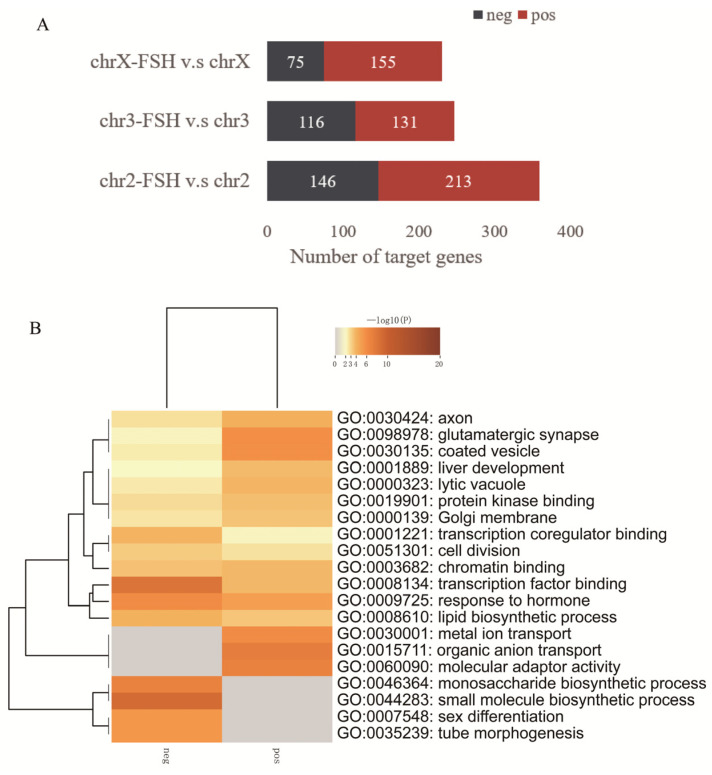
Positive and negative target gene results of massively parallel CRISPR-Cas9 screening. (**A**) The number of positive selection (pos) and negative selection (neg) target genes in the same chromosome treatment group compared with the control group; (**B**) heatmap of the top 20 enrichment clusters of positive and negative selection target genes.

**Figure 6 animals-14-00898-f006:**
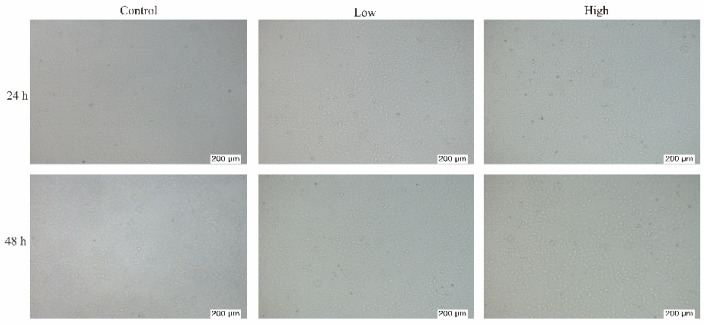
Microscopic examination of GCs’ growth with different doses of FSH. In the control group, no FSH was added to the medium; low dose means that 10 ng/mL FSH was added to the medium; and high dose means that 100 ng/mL FSH was added to the medium. Scale bars are 200 μm.

**Figure 7 animals-14-00898-f007:**
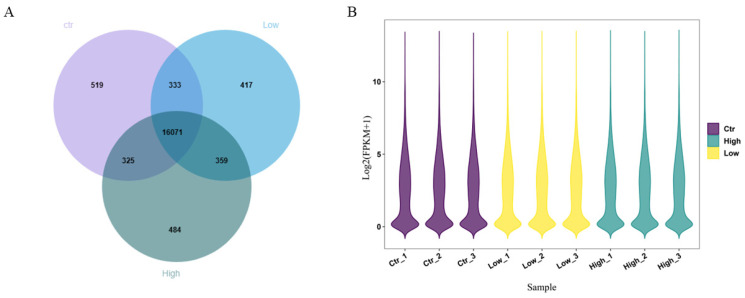
Characteristics of gene expression in GCs cultured in different doses of FSH in vitro. (**A**) Venn diagram showing the number of genes specifically expressed and co-expressed among the three groups. (**B**) Violin plot showing the expression of genes in GCs. FPKM: fragments per kilobase million.

**Figure 8 animals-14-00898-f008:**
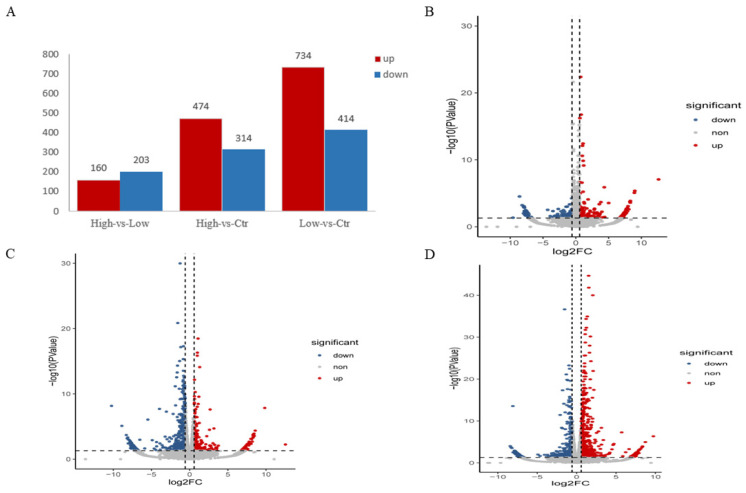
Changes in the expression profile of genes in GCs. (**A**), Number of different genes in each group of GCs; (**B**–**D**), volcano plot indicating up- and down-regulated genes of GCs in different FSH treatment groups (high versus low, high versus control, and low versus control), respectively.

**Figure 9 animals-14-00898-f009:**
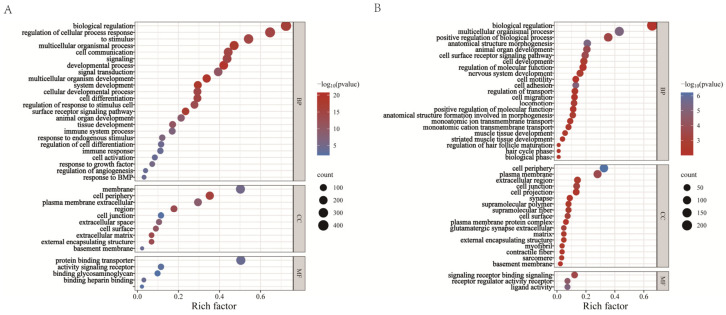
Gene ontology (GO) analysis of differentially expressed genes in different FSH treatment groups: (**A**) up-regulated genes; (**B**) down-regulated genes.

**Figure 10 animals-14-00898-f010:**
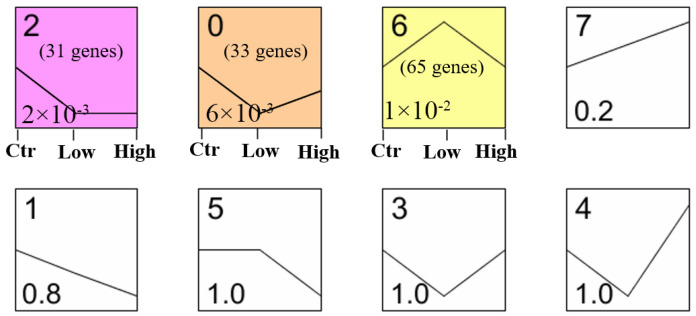
Eight profiles of positive–negative target genes with unique expression alterations over control (control group, 0 ng/mL FSH), low (low-dose group, 10 ng/mL FSH) and high (high-dose group, 100 ng/mL). The profile number is shown at the top left corner of each square. The *p*-value in each profile is shown at the bottom left corner of each square. Colored profiles have a statistically significant number of genes assigned, meanwhile the colors are randomly assigned.

**Table 1 animals-14-00898-t001:** Primer for amplifying the knockout cell library.

Primer Name	Primer Sequence (5′-3′)
sgRNA-F	AATTAATTTGACTGTAAACACAA
sgRNA-R	TTCAAGTTGATAACGGACT

**Table 2 animals-14-00898-t002:** CRISPR-Cas9 sgRNA library coverage and homogeneity.

Chr	sgRNA Sequencing ^1^	Number of sgRNAs ^2^	Number of Coding Genes ^3^	Coverage Number ^4^	Coverage ^5^	Homogeneity ^6^
chr2	13,750,412	5612	1433	5611	99.98%	11.2
chr3	15,763,418	7750	1990	7749	99.98%	4.1
chrX	14,493,832	3095	812	3095	100.00%	4.5

^1^ sgRNA sequencing: the number of reads in the sgRNA NGS sequencing. ^2^ Number of sgRNAs: the total number of different sgRNA sequences designed to target specific genes in different chromosomes of CRISPR-Cas9 experiments. ^3^ Number of coding genes: the number of target genes corresponding to sgRNA. ^4^ Coverage number: the number of times each target site was covered by a separate sgRNA sequence in the constructed sgRNA library. The higher the coverage, the greater the number of sgRNA sequences on each target site, thereby increasing the probability of knocking out each target site. ^5^ Coverage: the number of sgRNAs detected by NGS divided by the number of sgRNAs in the library’s theoretical list. ^6^ Homogeneity: the ratio of the minimum total number of reads in the highest 10% of reads to the maximum total number of reads in the lowest 10% should be less than 15.

**Table 3 animals-14-00898-t003:** KEGG pathway of up-regulated genes and down-regulated genes.

Type	KEGG_Name	KEGG_id	Adjusted *p*_Value	Gene_Count
all-up_DEGs	TNF signaling pathway	KEGG:04668	0.001371046	106
	PI3K-Akt signaling pathway	KEGG:04151	0.006758773	306
	ECM–receptor interaction	KEGG:04512	0.022827729	82
all-down_DEGs	Arrhythmogenic right ventricular cardiomyopathy	KEGG:05412	0.002590425	72
	Neuroactive ligand–receptor interaction	KEGG:04080	0.010170701	283
	cAMP signaling pathway	KEGG:04024	0.04874159	203

## Data Availability

Data are contained within the article and Appendix A.

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
