# Peer review of "Massively Parallel CRISPR-Cas9 Knockout Screening in Sheep Granulosa Cells for FSH Response Genes"

_animals, 2024, doi:10.3390/ani14060898_

Round 1
Reviewer 1 Report
Comments and Suggestions for Authors
In this manuscript, authors investigated the regulation of follicle-stimulating hormone (FSH) on granulosa cells (GCs) by exploring the optimal dose of FSH on the promotion of GCs, designing CRISPR-Cas9 knockout library on Chr 2, 3, and X to identify FSH-regulated genes under different doses. Overall, this manuscript provided detailed protocols, sound quality control, solid experimental results, and proper data interpretation and conclusions. However, there are a few comments and concerns I may have for this manuscript before its publication:
1. Reference 1 was not cited properly, there are two authors' names missing and labeled as “null”.
2. The author mentioned in lines 308-310 that “the sequencing reads have 50% of the reads that can not detect the sgRNA sequences, and the actual sgRNA sequence alignment rate was 39.14%, which was within the normal range.”, authors need to further provide literature references or explanation about why the 39% is normal with a waste of 61% reads.
3. Figure 6 is hard to visualize, authors may consider enlarging the graphs and increasing the resolution, or providing a quantitative graph for better comparison.
4. For section 3.5, the Authors need to add comments on why they chose a 100 ng/ml dose of FSH in the RNA-seq study. Based on Figure 1, 100ng/ml does not show a significantly different performance than the other concentrations. With 100ng/ml dose concentration, FSH still shows an improvement in proliferation in GCs starting day 4, compared to the control group. In section 3.5.1, the authors stated that the high-dose group showed an occurrence of cell death. However, based on Figure 1, there seems no significant difference between the control group and the high-dose group in cell viability. It is less meaningful to research the dose-dependent genes under the “best dose” and “ok dose”. It would be much more informative if the high-dose group was shown to be harmful to the GCs.
Author Response
Thank you for your kindly comments on our manuscript entitled " Massively CRISPR-Cas9 Knockout Screening in Sheep Granu-losa Cells for FSH Response Genes " (ID: animals-2853295). Those comments are very helpful for revising and improving our paper, as well as the important guiding significance to research. We have studied the comments carefully and made corrections which we hope meet with approval. The main corrections are in the manuscript and the responds to comment as follows:
Question1: Reference 1 was not cited properly, there are two authors' names missing and labeled as “null”.
Response1: I have modified it to: Liu, Z.; Fu, S.; He, X.; Dai, L.; Liu, X.; Narisu; Shi, C.; Gu, M.; Wang, Y.; Manda; Guo, L.; Bao, Y.; Baiyinbatu; Chang, C.; Liu, Y.; Zhang, W. Integrated Multi-Tissue Transcriptome Profiling Characterizes the Genetic Basis and Biomarkers Affecting Reproduction in Sheep (Ovis Aries). Genes (Basel) 2023, 14, 1881, doi:10.3390/genes14101881.
Question2: The author mentioned in lines 308-310 that “the sequencing reads have 50% of the reads that can not detect the sgRNA sequences, and the actual sgRNA sequence alignment rate was 39.14%, which was within the normal range.”, authors need to further provide literature references or explanation about why the 39% is normal with a waste of 61% reads
Response2: Thank you for your question. In genome-wide CRISPR-Cas9 applications, PCR next-generation sequencing (Illumina PE150) is often used to identify DNA cleavage and repair caused by sgRNA-guided CRISPR-Cas9 system, such as insertion and deletion (indels) detection. In this study, the size of the target sgRNA fragment is 250 bp, which means that in the double-end sequencing of PE150, the reads at both ends are not sufficient to merge and cover the entire fragment, but there may be some overlap, the sgRNA sequencing alignment rate was 39.14 %, and 60.86 % of them were not necessarily aligned. There may be the following reasons: 1) CRISPR-Cas9 may lead to off-target effect, that is, sgRNA randomly targets non-specific sites in the genome for editing. In this case, the sgRNA that plays a design role does not necessarily have the same high affinity as the off-target region, resulting in the location of some sgRNAs not being edited, so they cannot be detected in the sequencing data [1];
2) Some reads contain more N.We set strict comparison conditions when comparing, and we do not allow a base mismatch, and remove the N-containing reads, which does not affect the final effective clone, but the alignment rate will decrease.
3) In the model organisms, because their reference was usually of high quality, like the human genome-wide CRISPR screen [2]. However, for some non-model organisms the alignment rate may be lower.
Question3: Figure 6 is hard to visualize, authors may consider enlarging the graphs and increasing the resolution, or providing a quantitative graph for better comparison.
Response3: Thank you for your advice. I modified the picture to a high-resolution tiff format through Adobe Illustor software, but when inserting the manuscript, I always reported an error and showed insufficient memory, resulting in insertion failure. Therefore, I will upload high-definition images to the system, please check
Question4: For section 3.5, the Authors need to add comments on why they chose a 100 ng/ml dose of FSH in the RNA-seq study. Based on Figure 1, 100ng/ml does not show a significantly different performance than the other concentrations. With 100ng/ml dose concentration, FSH still shows an improvement in proliferation in GCs starting day 4, compared to the control group. In section 3.5.1, the authors stated that the high-dose group showed an occurrence of cell death. However, based on Figure 1, there seems no significant difference between the control group and the high-dose group in cell viability. It is less meaningful to research the dose-dependent genes under the “best dose” and “ok dose”. It would be much more informative if the high-dose group was shown to be harmful to the GCs.
Response4: The reasons for selecting a dose of 100 ng / ml FSH in RNA-seq studies are as follows : 1) according to the results of Figure 1, the addition of 100 ng/mL FSH with the exception of 10 ng/mL did not significantly differ compared to other concentrations, but the proliferation ability of GCs decreased from day 2 to day 3 and day 6. At the same time, we combined the published papers, and selected 100 ng/mL FSH as a simulated high FSH dose in vivo. As Sirotkin AV et al. has investigated that FSH hormone can directly regulate the ovarian mTOR/SIRT1 system (which can mediate the effect of hormone modulators on the ovaries), when added to 100 ng/mL, the expression of SIRT1 in GCs decreased [3].
2) This study considers both low-dose and high-dose responses were considered when evaluating hormone effects. Although the 100 ng/mL dose did not show a significant differences in the proliferation ability, it also can be seen from the number of differentially expressed genes and the functions performed by transcriptome analysis that it may reveal other regulatory mechanisms other than cell proliferation at the transcriptional level.
3) Focus on potential negative feedback or cellular stress response: GCs were treated with high doses of FSH to assume that it may activate the negative feedback loop or stress response in cells, which may be beneficial to explore the physiological adaptation mechanism of granulosa cells when FSH stimulation is excessive [4].
4) Ensure sufficient signals to induce changes in gene expression : High concentrations of FSH are used to ensure that all potential effects of aging or apoptosis-related gene expression patterns can be detected in transcriptome data, and these signals need to be strong enough to cause changes in upstream transcriptional regulation.
- Shalem, O.; Sanjana, N.E.; Zhang, F. High-Throughput Functional Genomics Using CRISPR–Cas9. Nat Rev Genet 2015, 16, 299–311, doi:10.1038/nrg3899.
- Toledo, C.M.; Ding, Y.; Hoellerbauer, P.; Davis, R.J.; Basom, R.; Girard, E.J.; Lee, E.; Corrin, P.; Hart, T.; Bolouri, H.; et al. Genome-Wide CRISPR-Cas9 Screens Reveal Loss of Redundancy between PKMYT1 and WEE1 in Glioblastoma Stem-like Cells. Cell Rep 2015, 13, 2425–2439, doi:10.1016/j.celrep.2015.11.021.
- Sirotkin, A.V.; Dekanová, P.; Harrath, A.H. FSH, Oxytocin and IGF-I Regulate the Expression of Sirtuin 1 in Porcine Ovarian Granulosa Cells. Physiol Res 2020, 69, 461–466, doi:10.33549/physiolres.934424.
- Behl, R.; Pandey, R.S. FSH Induced Stimulation of Catalase Activity in Goat Granulosa Cells in Vitro. Anim Reprod Sci 2002, 70, 215–221, doi:10.1016/s0378-4320(02)00006-4.
Reviewer 2 Report
Comments and Suggestions for Authors
1. General comments
This manuscript reports the comprehensive identification of FSH-responsive genes in sheep granulosa cells through combining CRISPR-Cas9 knockout screening and transcriptome analyses. It is of interest to combine two approaches to identify novel FSH-responsive genes or pathways that were unforeseen by previous studies. Nonetheless, this manuscript lacks a few things that should be addressed. In order to minimize the inclusion of previously identified genes, authors used the ch2, ch3, chX-specific CRISPR-Cas9 libraries which will be increase the possibility or capacity to identify the novel genes. But, As it is the first study in sheep cells, the CRISPR-Cas9 lib covering the whole sheep genome can be applied, and results can be compared to prove the hypothesis of this study. From the result, it is hard to figure out whether targeting specific chromosomes will perform better than targeting the whole genome.
2. Specific comments
(1) Each figure lacks a figure legend. The detailed caption should be included.
(2) In the MM section, the construction of CRISPR-Cas9 libraries for Ch 2, Ch 3, and Ch X should be described in more detail. Target genes from each library should be included in supplementary materials.
(3) The lists of genes that were identified from the chromosome-specific CRISPR-Cas9 libraries should be presented with genomic localization information.
(4) Line 194-199. For the selection experiment of CRISPR-Cas9 transduction followed by FSH stimulation, the description of the experimental procedures should be more in detail. After transduction and selection of CRISPR-Cas9 transduced granulosa cells, authors divided them into two groups, i.e., the FSH-stimulated group vs no FSH-group as the control.
(5) In Table 2, please include definition of the terms, such as coverage number, Coverage, and homogenesity. Or, you can include them in results section.
(6) In Figure 3. Scale bar is not clear. Needs to be improved.
(7) In Figure 5, what are the negative-selected genes and positive-selected genes? Please give them a definition in the text. Also, the gene lists should be included in supplementary tables.
(8) In Figure 6. The scale bar is not clear. Needs to be improved.
(9) DEG lists from transcriptome analyses should be included in supplementary tables.
(10) In Figure 10. Please give a detailed description of each subfigure.
Author Response
Thank you for your kindly comments on our manuscript entitled " Massively CRISPR-Cas9 Knockout Screening in Sheep Granu-losa Cells for FSH Response Genes " (ID: animals-2853295). Those comments are very helpful for revising and improving our paper, as well as the important guiding significance to research. We have studied the comments carefully and made corrections which we hope meet with approval. The main corrections are in the manuscript and the responds to comment as follows:
Question 1: Each figure lacks a figure legend. The detailed caption should be included.
Response 1: Thank you for your suggestions. Each figure have added a figure legend in the lastest manucript.
Question 2: In the MM section, the construction of CRISPR-Cas9 libraries for Ch 2, Ch 3, and Ch X should be described in more detail. Target genes from each library should be included in supplementary materials.
Response 2: Thank you for your suggestions. I have revised them in the lastest manuscript and added them in the Table S1.
The construction method of CRISPR-Cas9 library as follows :
1)Design sgRNA sequence: Firstly, the sgRNA sequences of protein-coding genes on the sheep chromosomes 2,3 and X were designed, utilizing the sgRNACas9 online software to search for sgRNAs with higher specificity as candidate sgRNA. The length of sgRNA was designed to be 20 nt.
2) Synthesis of sgRNA sequences: After design completion, the corresponding sgRNA sequences were synthesized using the Syno ® 3.0 technology platform, and the synthesized oligonucleotide probes were subjected to PCR amplification with high-fidelity enzymes. Amplification products were separated by 2 % agarose gel elec-trophoresis and recovered.
3) Cloning into the vector : The amplified product was cloned into the CRISPR _ Cas9 lentiviral expression vector Lenti CRISPR.v2 vector to form a linearized vector.
4) Construction of a library : Using Gibson Assembly ® MasterMix, an enzyme for DNA fragment assembly, multiple DNA fragments are smoothly spliced together in one step. The recovered library fragment product was ligated with linearized vector by Gibson Assembly ® MasterMix, and the obtained Gibson assembly product was trans-formed into Trans1 T1 competent cells. Then the sgRNA plasmid library was further obtained by plasmid extraction kit extraction.
5) Packaging and transfection : The lentivirus-related packaging plasmid and the recombinant plasmid carrying the sgRNA library were transfected into 293T cells. The constructed sgRNA library vector was packaged into the lentivirus, and then 293T cells produced lentivirus carrying sgRNA. The lentivirus was used to infect sheep granulosa cells for gene editing experiments.
Question 3: The lists of genes that were identified from the chromosome-specific CRISPR-Cas9 libraries should be presented with genomic localization information.
Response 3: Thank you for your suggestions. I 've provided detailed gene list information in Table S1. Chromosome-specific gene list _CRISPR-Cas9 libraries
Question 4: Line 194-199. For the selection experiment of CRISPR-Cas9 transduction followed by FSH stimulation, the description of the experimental procedures should be more in detail. After transduction and selection of CRISPR-Cas9 transduced granulosa cells, authors divided them into two groups, i.e., the FSH-stimulated group vs no FSH-group as the control.
Response 4: In the lastest manuscript, I have described the experimental procedures in part 2.6.
Granulosa Cells knockout was performed on sheep chromosome 2, 3 and X libraries. After GCs were infected with each library virus, we divided the cell library into two groups. The knockout cell library obtained by only 1 ng/mL puromycin screening was the control group (without FSH stimulation,), and the FSH stimulation group with 10 ng/mL FSH.
Question 5: In Table 2, please include definition of the terms, such as coverage number, Coverage, and homogenesity. Or, you can include them in results section.
Response 5: I ' ve added specific noun explanations in the form of annotations at the bottom of Table 2.
Question 6: In Figure 3. Scale bar is not clear. Needs to be improved.
Response 6: Thank you for your advice. I re-improved the clarity of the picture and uploaded it to the system
Question 7: In Figure 5, what are the negative-selected genes and positive-selected genes? Please give them a definition in the text. Also, the gene lists should be included in supplementary tables.
Response 7: I defined positive and negative-selected genes in MM 2.6 part,. And I've provided detailed gene list information in Table S3. Target genes list_negative and positive-selected.
Positive and negative selection were based on the phenotype of interest and available selection pressures for the screen. Positive selection screening is to apply certain screening pressure to the cell library that has successfully integrated sgRNAs, and the number of sgRNAs associated with phenotype increases relative to other sgRNAs. Depending on the enrichment of sgrna to obtain genetic perturbations that produce screening phenotypes due to cell proliferation, resistance genes can be screened. Negative selection screening involves the depletion of phenotypic corresponding sgRNAs due to cell death. However, for mass screening, the phenotype of interest does not lead to cell proliferation or cell death, so phenotypically associated sgRNAs are not enriched or depleted, and genes necessary for cell survival can be screened.
Joung, J.; Konermann, S.; Gootenberg, J.S.; Abudayyeh, O.O.; Platt, R.J.; Brigham, M.D.; Sanjana, N.E.; Zhang, F. Genome-Scale CRISPR-Cas9 Knockout and Transcriptional Activation Screening. Nat Protoc 2017, 12, 828–863, doi:10.1038/nprot.2017.016.
Question 8: In Figure 6. The scale bar is not clear. Needs to be improved.
Response 8: Thank you for your advice. I modified the picture to a high-resolution tiff format through Adobe Illustor software, but when inserting the manuscript, I always reported an error and showed insufficient memory, resulting in insertion failure. Therefore, I will upload high-definition images to the system, please check.
Question 9: DEG lists from transcriptome analyses should be included in supplementary tables.
Response 9. I 've provided detailed gene list information in Table S6.DEG lists_GCs transcriptome analyses.
Question 10: In Figure 10. Please give a detailed description of each subfigure.
Response 10. I added a detailed description to Figure 10 in the lastest manicript.
Reviewer 3 Report
Comments and Suggestions for Authors
The authors investigated the role of FSH in regulating granulosa cells. They developed a CRISPR-Cas9 knockout library targeting genes on chromosomes 2, 3, and the X chromosomes of sheep, including an ovarian GC knockout cell library. These findings uncovered regulatory factors, offering insights for further research in sheep reproduction. I have following questions/comments for the authors:
1. It would be good to comment why higher concentration of FSH (15, 20 or 30ng/ml) were not able to increase the proliferation of GC as good as 10ng/ml in Fig. 1?
2. Please elaborate on what is meant by " The number of sgRNAs was 44,007,662 reads, corresponding to 265 18,207 sgRNAs " in line 265.
3. Lines 273-280 should belong to methods section.
4. Can authors comment on why the percentage of unmapped reads were so high?
5. What does authors mean by positive screening genes and negative screening genes? It should be made clear in the 3.4.2 section
6. In line 352, what are the three levels of FSH? What is the high/low dose (value)? It should be clearly mentioned.
7. In fig 7a, can authors shed light on the 325 genes expressed by control and high dose and not the low dose? What can be interpreted by that?
8. In fig 8, what does the volcano plot in b, c and d depict? How are they different?
The legends for all the figures need to be elaborate.
Comments on the Quality of English Language
Need improvement
Author Response
Thank you for your kindly comments on our manuscript entitled " Massively CRISPR-Cas9 Knockout Screening in Sheep Granu-losa Cells for FSH Response Genes " (ID: animals-2853295). Those comments are very helpful for revising and improving our paper, as well as the important guiding significance to research. We have studied the comments carefully and made corrections which we hope meet with approval. The main corrections are in the manuscript and the responds to comment as follows:
Question 1: 1. It would be good to comment why higher concentration of FSH (15, 20 or 30ng/ml) were not able to increase the proliferation of GC as good as 10ng/ml in Fig. 1?
Response 1: Thank you for your suggestions. The response of cells to FSH, which is affected by the concentration of FSH, may exhibit different sensitivity or ' dose response ' characteristics for a variety of reasons. Within a certain concentration range, FSH will stimulate the proliferation of granulosa cells. However, too high concentration of FSH does not necessarily continue to increase cell proliferation, and may even lead to a decrease in proliferation rate. For example, high concentrations of FSH can trigger the internalization and degradation of FSH receptors on the cell surface, resulting in a decrease in the number of receptors, thereby reducing the cell 's response to FSH ; when the concentration of FSH in cell culture is high, it may cause changes in important growth factors, hormones and intercellular communication in the culture medium, resulting in the inability to maintain the environment necessary to support granulosa cell proliferation ; with the increase of cell density, no matter what the concentration of FSH is, these limiting factors may become the constraints of cell proliferation. At present, no more specific effect of a certain FSH dose on the growth of sheep granulosa cells has been found. In this study, CCK-8 reagent showed that the cell proliferation rate was significantly increased by adding 10 ng/mL FSH, and it was preliminarily speculated that this dose provided suitable biological activity for granulosa cells. The effect of FSH dose on granulosa cell proliferation may be affected by many external factors such as reagent brands, which will be further explored in the future.
Question 2: Please elaborate on what is meant by " The number of sgRNAs was 44,007,662 reads, corresponding to 265 18,207 sgRNAs " in line 265.
Response 2: Thank you for your question. I have explained these nouns in Part 3.2. Some of the numbers have been corrected in the latest manuscript because of my carelessness and filling errors. I apologize again for causing you confusion.
Question 3: Lines 273-280 should belong to methods section.
Response 3: Thank you for your suggestions. I have revised them in the lasted manuscript.
Question 4: Can authors comment on why the percentage of unmapped reads were so high?
Response 4: Thank you very much for your interest in my research. Here I make a shallow explanation, I hope my explanation for you to solve the puzzle. In genome-wide CRISPR-Cas9 applications, PCR next-generation sequencing (Illumina PE150) is often used to identify DNA cleavage and repair caused by sgRNA-guided CRISPR-Cas9 system, such as insertion and deletion (indels) detection. In this study, the size of the target sgRNA fragment is 250 bp, which means that in the double-end sequencing of PE150, the reads at both ends are not sufficient to merge and cover the entire fragment, but there may be some overlap, the sgRNA sequencing alignment rate was 39.14 %, and 60.86 % of them were not necessarily aligned. There may be the following reasons: 1) CRISPR-Cas9 may lead to off-target effect, that is, sgRNA randomly targets non-specific sites in the genome for editing. In this case, the sgRNA that plays a design role does not necessarily have the same high affinity as the off-target region, resulting in the location of some sgRNAs not being edited, so they cannot be detected in the sequencing data [1];
2) Some reads contain more N.We set strict comparison conditions when comparing, and we do not allow a base mismatch, and remove the N-containing reads, which does not affect the final effective clone, but the alignment rate will decrease.
3) In the model organisms, because their reference was usually of high quality, like the human genome-wide CRISPR screen [2]. However, for some non-model organisms the alignment rate may be lower.
Question 5: What does authors mean by positive screening genes and negative screening genes? It should be made clear in the 3.4.2 section
Response 5: I have explained them in part 2.6 because of firstly occurance: Positive and negative selection were based on the phenotype of interest and available selection pressures for the screen. Positive selection is to exert a certain screening pressure on the cell library that has successfully integrated sgRNAs, and the number of phenotype-related sgRNAs is increased relative to the rest of the sgRNAs. Depending on the enrichment of sgRNA to obtain genetic perturbations that produce screening phenotypes due to cell proliferation, which can screen resistance genes. Negative selec-tion involves the depletion of phenotypic corresponding sgRNAs due to cell death, and the genes necessary for cell survival can be screened [5].
Question 6: In line 352, what are the three levels of FSH? What is the high/low dose (value)? It should be clearly mentioned.
Response 6: Thank you for your advice. I have added it in part 3.5.1.
Question 7: In fig 7a, can authors shed light on the 325 genes expressed by control and high dose and not the low dose? What can be interpreted by that?
Response 7: Thank you for your question. I ' m very sorry, I ' m very serious to find your question, but did not find the specific number of 325. But I generally understand the question you 're asking. The reason for the expression of genes in the control and high-dose rather than low-dose treatment may reveal a dose effect, in which the difference in gene expression between the control group ( no treatment or conventional treatment ) and the high-dose treatment group, and these differences did not appear or were not obvious at low doses. This indicates that the expression of these genes will be significantly regulated only when a certain threshold dose is reached. It may also be due to the potential molecular and cellular mechanisms of this dose-dependent regulation. For example, certain signaling pathways may not be fully activated at low doses, but are over-activated at high doses, resulting in differentially expressed genes in the regulatory network.
If I understand that there is a deviation, please criticize, thank you very much.
Question 8: In fig 8, what does the volcano plot in b, c and d depict? How are they different? Response 8: The Figure 8(B) were three volcanic diagrams drawn by analyzing the differential expression between the three groups indicating up and down regulated genes of GCs in different FSH treatment groups (High versus Low, High versus Ctr and Low versus Ctr), respectively.
Round 2
Reviewer 2 Report
Comments and Suggestions for Authors
The revision has been made well and is satisfactory. Therefore, it is acceptable for the journal.
Reviewer 3 Report
Comments and Suggestions for Authors
I am satisfied with the comments from the authors.
Comments on the Quality of English Languageneeds improvement